# Adapting Reinforcement Learning for Path Planning in Constrained Parking Scenarios

## Abstract

Real-time path planning in constrained environments remains a fundamental challenge for autonomous systems. Traditional classical planners, while effective under perfect perception assumptions, are often sensitive to real-world perception constraints and rely on online search procedures that incur high computational costs. In complex surroundings, this renders real-time deployment prohibitive. To overcome these limitations, we introduce a Deep Reinforcement Learning (DRL) framework for real-time path planning in parking scenarios. In particular, we focus on challenging scenes with tight spaces that require a high number of reversal maneuvers and adjustments. Unlike classical planners, our solution does not require ideal and structured perception, and in principle, could avoid the need for additional modules such as localization and tracking, resulting in a simpler and more practical implementation. Also, at test time, the policy generates actions through a single forward pass at each step, which is lightweight enough for real-time deployment. The task is formulated as a sequential decision-making problem grounded in a bicycle model dynamics, enabling the agent to directly learn navigation policies that respect vehicle kinematics and environmental constraints in the closed-loop setting. A new benchmark is developed to support both training and evaluation, capturing diverse and challenging scenarios. Our approach achieves state-of-the-art success rates and efficiency, surpassing classical planner baselines by **+96%** in success rate and **+52%** in efficiency. Furthermore, we release our benchmark as an open-source resource for the community to foster future research in autonomous systems.

## 1 Introduction

Classical path planners, such as Hybrid A* (Dolgov et al., 2008), have long been widely used in autonomous systems to compute feasible trajectories. Given precise and complete perception observations, these methods can generate near-optimal[1] paths for tasks such as autonomous parking. However, in real-world scenarios, perception is inherently uncertain and often occluded in tight spaces, leading to brittle plans. For instance, as shown in Figure 1, paths computed under partial observability may result in unavoidable collisions. Moreover, classical planners do not retain prior knowledge beyond simple heuristics, causing them to repeatedly search for solutions online. This introduces significant risk of exceeding onboard computational limits, particularly in complex surroundings. Finally, the integration of classical planners into a full autonomy stack requires additional modules—such as localization and path tracking—that themselves introduce uncertainty and compounding errors across the system. These limitations motivate us to explore alternative approaches for solving the path planning task in constrained environments.

Recent advances in machine learning have inspired AI-based solutions for path planning (Jiang et al., 2023; Lazzaroni et al., 2023; Chi et al., 2023; Yang et al., 2024; Zheng et al., 2025). Broadly, these approaches can be categorized into open-loop and closed-loop training paradigms. Open-loop methods, such as supervised imitation learning (Ahn et al., 2022), are simple to implement but prone to distribution shift, limiting their generalizability to unseen scenarios. They also do not explicitly en-

---

[1]In principle, Hybrid A* can recover the globally optimal path if allowed sufficient search time. However, in practice it is often combined with heuristic shortcuts such as Reeds-Shepp curves to accelerate search, which yields a solution but not truly the optimal one.

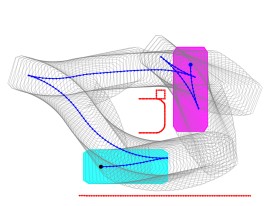 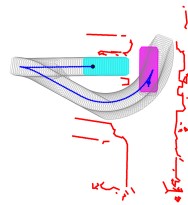

(a) Dead-end parking  (b) Corridor parking

Figure 1: Non-optimal paths generated by the Hybrid A* planner in constrained parking environments. The start pose is denoted by magenta rounded rectangles, and the target pose by cyan rounded rectangles. Planned paths are shown in blue, with shaded gray regions indicating the space occupied by the vehicle along its intermediate poses. Due to partial observability, the current solutions for both cases are likely to result in collisions (obstacles are denoted by solid red lines).

force that predicted paths are dynamically feasible and trackable, particularly for challenging parking maneuvers. Closed-loop training, by contrast, directly accounts for sequential decision-making and vehicle feedback, thereby improving robustness and generalization. Yet, closed-loop learning for constrained path planning remains underexplored, especially in the autonomous driving field, largely due to (1) the absence of a standardized benchmark that reflects the tight spatial conditions encountered in practice, and (2) the challenge of designing RL pipelines, where the reward function and training strategy must be carefully tuned.

In this work, we propose a Reinforcement Learning-based planner to address the limitations of both classical and existing AI-driven approaches. In particular, we formulate path planning as a sequential decision-making problem under a bicycle model dynamics, enabling the planner to explicitly respect kinematic constraints. We develop our own RL training strategy with curriculum learning and balance effective exploration with precise vehicle control by adopting an action-chunking mechanism (Li et al., 2025). To support both training and evaluation, we construct a benchmark, named as **ParkBench**, tailored to constrained scenarios and build a simulation environment that leverages this benchmark for closed-loop interactions. Our approach achieves the state-of-the-art performance on the proposed benchmark, significantly surpassing classical planner baselines in a large margin.

Overall, our key contributions are summarized below:

- We formulate the path planning problem as a reinforcement learning task grounded in a bicycle model dynamics, and provide a detailed design methodology.

- We propose an action chunking wrapper as a mechanism to reconcile accurate movement control with effective RL exploration.

- We achieve state-of-the-art results on constrained path planning and release our benchmark, **ParkBench**, as an open-source dataset to foster future research in this direction.

## 2 RELATED WORKS

### 2.1 CLASSICAL PATH PLANNERS

Classical planners form the foundation of autonomous navigation and parking systems. Among them, Hybrid A* is one of the most widely adopted algorithms, combining grid-based search with continuous state interpolation to ensure feasible trajectories under vehicle kinematics. In principle, Hybrid A* can recover the optimal path with sufficient search time, but in practice, it is accelerated through heuristic shortcuts such as Reeds-Shepp curves (Reeds & Shepp, 1990), yielding near-optimal solutions that are computationally tractable. In our work, we adopt a publicly available Hybrid A* implementation[2] (Sakai et al., 2018) as a strong baseline, ensuring a fair comparison between reinforcement learning-based and classical planning approaches.

---

[2]https://github.com/AtsushiSakai/PythonRobotics/tree/master/PathPlanning/HybridAStar.

## 2.2 AI-BASED PLANNING APPROACHES

Recent years have seen growing interest in leveraging learning-based methods for path planning. For instance, VAD (Jiang et al., 2023) predicts a sequence of future waypoints conditioned on scene context, achieving good performance on the nuScenes benchmark (Caesar et al., 2020). However, its training follows the open-loop paradigm and thus suffers from covariate shift, where compounding errors lead to distribution drift at test time, and they cannot guarantee dynamically trackable paths in complex maneuvers such as parking. Another family of supervised approaches leverages diffusion models for trajectory generation. Examples such as Diffusion Policy (Chi et al., 2023) demonstrate strong generative capabilities, but generated paths must be explicitly constrained to ensure trackability, and they require large-scale expert demonstrations for training.

In contrast, closed-loop reinforcement learning (RL) approaches train agents through trial-and-error interactions in simulated environments, directly accounting for sequential decision-making. While promising, existing RL studies (Lazzaroni et al., 2023; Al-Mousa et al., 2025) on parking remain limited by overly simplified environments and do not address the tight constrained spaces that characterize realistic parking scenarios. This gap highlights the need for more challenging benchmarks and robust learning methods that can generalize beyond toy settings.

Deep reinforcement learning (DRL) has been widely explored in mobile robot navigation (Zhu & Zhang, 2021). However, these works are not directly applicable to the parking task we study in this work. Most navigation methods (Pérez-D'Arpino et al., 2021; Ruan et al., 2019; Xu et al., 2022; Akmandor et al., 2022) assume differential-drive robots with highly flexible motion capabilities, whereas parking requires vehicle modeling governed by nonholonomic constraints such as the bicycle or the Ackermann-steering models. These kinematic models restrict maneuverability. A related study uses an RC-car platform and combines model-free and model-based RL for indoor navigation (Kahn et al., 2018). Despite these efforts, navigation goals are typically treated as waypoints without enforcing precise final orientation, while parking requires exact terminal conditions. To the best of our knowledge, few learning-based methods jointly consider these constraints, motivating the development of our RL-based parking planner.

## 2.3 COMBINING CLASSICAL AND LEARNING-BASED METHODS

Another active line of research integrates classical planners with machine learning techniques to combine the strengths of both paradigms. Recent works (e.g., (Shan et al., 2023; Jiang et al., 2025)) use learned models to guide search, accelerate tree expansion, or provide better heuristics for classical planners. These hybrid approaches hold promise for balancing efficiency and generalizability. However, such methods remain outside the scope of this work, as our focus is on demonstrating the viability of a purely reinforcement learning-based planner in constrained path planning scenarios. We view the integration of RL with classical heuristics as a valuable direction for future research.

## 2.4 PARKING EVALUATION BENCHMARK

To the best of the authors' knowledge, there are few practical benchmarks available for evaluating path planners, particularly in constrained parking scenarios. Among the limited existing attempts, the E2E Parking benchmark (Yang et al., 2024) leveraged CARLA (Dosovitskiy et al., 2017) to create a parking simulation environment, but its task setting is restricted to rear-in perpendicular parking in wide open spaces. Another notable effort is the TPCAP benchmark (Li et al., 2022), designed for an autonomous parking competition and consisting of 20 parking challenge cases. However, TPCAP represents obstacles as solid shapes and focuses solely on planning, which makes its formulation incompatible with existing autonomous driving pipelines. Moreover, the scenarios in TPCAP are overly simplified and not representative of realistic real-world conditions.

## 3 METHODOLOGY

In this section, we present a reinforcement learning (RL) framework for path planning in constrained parking scenarios, designed as a drop-in replacement for the Hybrid A* module in the autonomous driving pipeline. This design choice ensures compatibility with existing autonomy stacks and enables a fair comparison against a strong classical baseline as well. Our methodology is organized

into five components. We first formulate the parking problem as a sequential decision-making task under vehicle kinematics. Next, we describe the input representation, which mirrors Hybrid A* to maintain pipeline consistency, along with our strategy to address the resulting training challenges. We then introduce our benchmark and simulation environment, followed by detailed training objective and reward design. Finally, we introduce a plug-in action-chunking mechanism that balances exploration efficiency with maneuver precision.

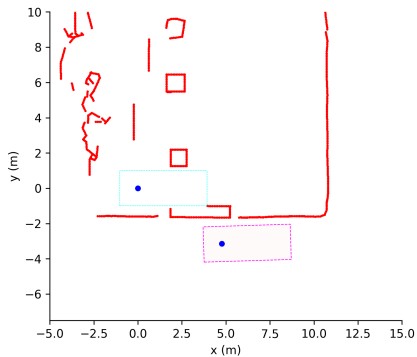 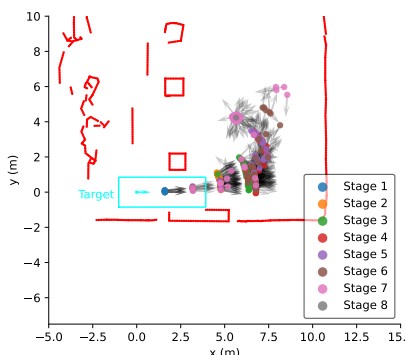

(a) One example of the non-feasible initial pose (blocked by a wall).

(b) Initial poses generated by our rollout method for scenario (a).

Figure 2: Challenges in spawning ego initial poses in the sparse obstacle representation environment and our rollout solution. Arrows in (b) represent the heading directions, respectively.

## 3.1 PROBLEM FORMULATION

We formulate the constrained path planning task as a sequential decision-making problem. The vehicle state is defined as $(x, y, \theta, \delta)$, representing the 2D position of the rear-axle center, the heading angle, and the front wheel steering angle, respectively. The environment provides obstacle information in the form of contour points' 2D coordinates, such as in lidar scans, of the type $(x^{obs,1}, y^{obs,1}, ..., x^{obs,N}, y^{obs,N})$, where $N$ is the maximum number of obstacle points considered. A target parking pose $(x^{goal}, y^{goal}, \theta^{goal})$ is also provided. Here, all the coordinates are expressed in world frame.

At each iteration, the simulator updates the pose of the vehicle by executing the selected 1-step control action through a kinematic bicycle model (Rajamani, 2006), a common abstraction for autonomous driving applications. This ensures that the learned policy respects nonholonomic constraints. The control space is discrete, and consists of two components $(\Delta s, \triangle \delta)$:

1. displacement along longitudinal axis $\Delta s \in \{+ds, -ds, 0\}$, representing forward, backward, or no motion with distance $ds > 0$ in meters,

2. front wheel steering change $\triangle \delta \in \{+d\delta, -d\delta, 0\}$, representing left, right, or no change in radians. When the vehicle does not move, left and right steering changes are possible.

The ego state is therefore updated via the discrete-space bicycle model as:

$$x_{k+1} = x_k + \Delta s_k \cdot \cos(\theta_k),$$
$$y_{k+1} = y_k + \Delta s_k \cdot \sin(\theta_k),$$
$$\theta_{k+1} = \theta_k + \Delta s / W_B \cdot \tan(\delta_k), \quad (1)$$
$$\delta_k = \delta_{k-1} + \Delta \delta_k,$$

where $W_B$ denotes the vehicle wheelbase in meters and subscript $k$ denotes the iteration. Iteratively applying these updates will produce the complete planned path (waypoint sequence).

The planning objective is to generate a feasible action sequence that drives the vehicle from its initial state to the target parking pose without collisions, while make sure the derived path is reasonable (we will quantify the quality of the path in subsection 3.4).

## 3.2 INPUT REPRESENTATION

To ensure fairness in comparison and maintain pipeline consistency, we design the input to our RL planner to match the information used by the Hybrid A* baseline. Specifically, the input includes the ego's current pose, the target pose, and obstacle contour features extracted from the environment. While this ensures comparability, it also introduces additional challenges for RL: unlike Hybrid A*, the RL training process must handle diverse ego initializations, and the sparse obstacle representation (given as obstacle contours) makes certain spawn positions particularly problematic. The key issue is that collision and feasibility checks cannot be reliably performed for the initial pose using only sparse contours, which can result in infeasible configurations such as the ego starting outside a wall or in positions with no valid path into the parking space (as shown in Figure 2a).

To address this challenge, we employ a roll-out function that gradually drives the ego away from the target pose using the bicycle model, with perturbations added to the heading for diversity. This procedure guarantees that the sampled initial poses are feasible, effectively reducing variance in training and improving convergence. An illustration of the sampled initial poses is shown in Figure 2b.

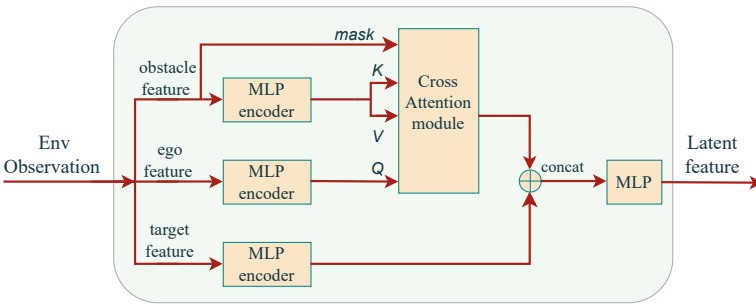

**Feature extractor module**

Figure 3: Our feature extractor architecture for vectorized environment observations.

The sparse contour representation of obstacles originates from the design of the existing pipeline, where Hybrid A* performs collision checks during its search from a given feasible starting pose to the target parking spot. Since Hybrid A* only requires obstacle boundaries for this purpose, obstacles are encoded as contours rather than dense occupancy maps or volumetric representations. To process this observation space effectively for the RL planner, we propose a feature extractor (Figure 3) that employs cross-attention to force the ego to attend to obstacle information. All coordinates are transformed into the ego-centric coordinate system at each time step (such that the ego is always located at the origin) and normalized before being passed to the feature extractor. We also impose a finite horizon range on the input to mimic the sensing limits of a perception module. This design choice makes the setting more realistic for deployment and could, in future work, help reduce reliance on additional downstream modules such as localization and path tracking, since the RL agent directly outputs control commands. It may also provide robustness to perception noise in the first frame and open the door to handling dynamic obstacles (e.g., moving vehicles or pedestrians). While these aspects are beyond the scope of this work, our input design highlights the potential of RL-based planners to integrate seamlessly into more complex real-world scenarios. It is worth mentioning that our feature extractor is intentionally lightweight to ensure feasibility for real-time, on-device deployment, which is a key requirement of practical parking systems.

## 3.3 BENCHMARK AND SIMULATION ENVIRONMENT

The missing of proper benchmarks for parking evaluation motivate the development of our benchmark, **ParkBench**, which is specifically tailored to constrained parking scenarios. Each scenario specifies the ego's initial pose, the target pose, and the positions of obstacles (contours) that define tight maneuvering spaces. Our current **ParkBench** includes 51 set of scenario layouts (all extracted from real-world dataset) for rear-in parking tasks, ranging from narrow aisles to occluded corner spots, reflecting the challenges of real-world parking. Detailed layouts are provided in Appendix G.

Based on this benchmark, we build a simulation environment that follows the Gym interface, ensuring compatibility with standard RL libraries. The environment is initialized by loading one of the benchmark scenarios, after which the RL agent can interact with it and evolve through sequential actions (see Figure 4 for an overview of our closed-loop method). Our simulator design is similar in spirit to (Scheel et al., 2022), which was developed for closed-loop training in autonomous driving. Note here, the environment state updates are computed under the bicycle model with the simplifying assumption that no dynamic obstacles are present, *i.e.*, only static obstacles are considered.

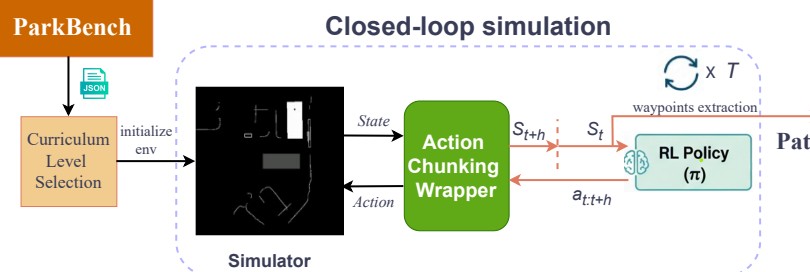

Figure 4: Overview of our closed-loop path generation method. The simulator is initialized with a realistic parking scenario, and the environment is iteratively updated based on the RL policy. This framework enables both training the policy and extracting planned paths during inference.

## 3.4 RL TRAINING WITH CURRICULUM LEARNING

Once the simulator is available, we can train the RL policy by interacting with the environment in a closed-loop manner. We adopt Stable Baselines3 (SB3) (Raffin et al., 2021) as the training framework and use Proximal Policy Optimization (PPO) (Schulman et al., 2017) as the base algorithm.

The key to successful RL training lies in the design of the reward function. However, designing dense rewards for parking tasks is challenging, and ill-defined reward functions often lead to unintended behaviors. To address this, we adopt a sparse reward formulation instead. The reward function consists of the following components:

- **Goal achievement**: a positive reward is given if the ego reaches the target pose within tolerance.

- **Collision penalty**: a negative reward is applied if the ego collides with any obstacle contour.

- **Out-of-bounds penalty**: a negative reward if it moves too far away from the valid maneuvering space.

- **Idle penalty**: a small negative reward to discourage the agent from remaining idle.

- **Direction-change penalty**: a small negative reward to penalize gear changes (switching between forward and backward) for smooth paths.

- **Time penalty**: a small negative reward applied at each step to incentivize faster completion.

Of course, sparse rewards will make it difficult for an RL agent to learn, especially in complicated tasks. To make training more effective, we integrate this sparse reward design with curriculum learning (Florensa et al., 2017) by gradually increasing scenario difficulty: starting from initial poses close to the target, then progressively moving further away from the target plus heading angle perturbation (refer Figure 2b). This progression helps the agent first acquire basic maneuvering skills before tackling the full complexity of constrained parking tasks. It also reduces unsafe or wasteful exploration: early stages restrict initial conditions to feasible neighborhoods (near the target with small heading perturbations), keeping rollouts within valid free space and mitigating collisions and feasibility violations. As the difficulty increases, the agent gradually expands its coverage while retaining a learned prior over valid configurations.

Table 1: Comparison on ParkBench. Best results are marked in **bold**. "CL" denotes whether curriculum learning is used, and "Chunking" denotes whether action chunking is used. PPO with curriculum training but without action chunking exhibits a large number of pivot points due to oscillatory behavior, which motivated the introduction of action chunking.

| Method | CL | Chunking | Succ. (%) ↑ | Time (s) ↓ | Dist. (m) ↓ | Pivot Points ↓ |
|---|---|---|---|---|---|---|
| Hybrid A* | ✗ | ✗ | 47.1 | 0.42 | 22.3 | **3.2** |
| PPO (Ours) | ✓ | ✗ | 62.7 | 0.72 | 21.7 | 53.4 |
| PPO (Ours) | ✓ | ✓ | **92.2** | **0.20** | **19.2** | 4.3 |

With this sparse reward design and curriculum learning strategy, policies can be trained end-to-end within our simulation environment, producing agents capable of executing collision-free parking maneuvers in tightly constrained spaces.

### 3.5 ACTION CHUNKING FOR EFFICIENT LEARNING

The default setting for RL algorithms is to select and execute one primitive action at a time. However, this setting is not well suited to parking tasks in constrained spaces. Training RL agents in such environments requires balancing the trade-off between exploration efficiency and precise movement control. Fine-grained primitive actions (*e.g.*, small steering adjustments) enable accurate maneuvering but make exploration highly inefficient due to long horizons. Conversely, coarse actions improve exploration efficiency but reduce maneuver precision, often leading to collisions.

To address this challenge, we adopt an action chunking mechanism, inspired by a recent work on Q-chunking (Li et al., 2025). In our formulation, a chunk corresponds to a short sequence of low-level control commands executed as a single macro-action. This reduces the effective planning horizon while preserving sufficient control fidelity, enabling efficient exploration without sacrificing maneuver precision. Different from the Q-chunking work, which introduces a modified Q-value function $Q(s_t, a_{t:t+h})$ where $h$ denotes the chunk length, and is therefore restricted to Q-learning–based methods, our formulation is more general. In particular, our action chunking mechanism is implemented as an environment wrapper, allowing it to be seamlessly applied to any RL algorithm without modifying the underlying training objective. The pseudocode for our training pipeline is show in Appendix F (Algorithm 1).

## 4 EXPERIMENTS

In this section, we will evaluate our training methodology on the **ParkBench** benchmark and compare it with both the classical and standard RL baselines.

### 4.1 EVALUATION SETUP

We first train our RL approach with action chunking ($h = 4$) as well as the standard RL baseline following the same training strategy described in 3.4. The detailed reward values and curriculum learning stages are provided as follows:

**Reward values:** The reward function is defined as:

$$r = R_{\mathrm{g}} \cdot \mathbb{1}_{\mathrm{goal}} + R_{\mathrm{c}} \cdot \mathbb{1}_{\mathrm{collision}} + R_{\mathrm{out}} \cdot \mathbb{1}_{\mathrm{out\_of\_bounds}} + R_{\mathrm{gear}} \cdot \mathbb{1}_{\mathrm{direction\_change}} + R_{\mathrm{idle}} \cdot \mathbb{1}_{\mathrm{idle}} + R_{\mathrm{time}}, \quad (2)$$

where $R_{\mathrm{g}} = 3, R_{\mathrm{c}} = -3, R_{\mathrm{out}} = -3, R_{\mathrm{gear}} = -0.01, R_{\mathrm{idle}} = -0.2, R_{\mathrm{time}} = -0.01$, and we use $\mathbb{1}_{condition}$ to denote the indicator of the condition is reached, which equals 1 when condition holds and 0 otherwise. The tolerance for reaching the target pose is set as 0.2 meter (with respect to the geometric center) and $\pm 3$ degrees in heading difference.

**Curriculum learning stages:** In this work, we define a multi-stage curriculum learning process. In particular, we set up 8 stages for the complete training iterations, the first 7 stages are illustrated in Figure 5 and the last stage uses the logged initial poses for the learning agent.

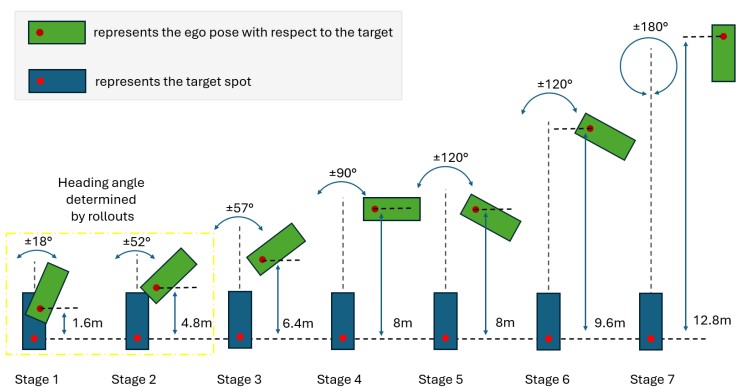

Figure 5: First seven stages in the curriculum learning. Stage 1 and 2 directly inherit the heading angle from rollout results. For other stages, the ego heading is reset to a collision-free angle sampled from the stage-specific range. All seven stages, the lateral offset is taken from the corresponding rollout result.

We evaluate our method on the **ParkBench** using the original logged ego pose as the starting point for each scenario, ensuring consistency across different planners. To assess performance, we report four key metrics: (1) **Success rate**: the fraction of cases where the ego successfully reaches the target pose within a tolerance on position and orientation; (2) **Planning time**: the average computation time required to generate a feasible trajectory; (3)**Travel distance**: the total path length of the executed trajectory, measuring efficiency; and (4) **Pivot points**: the number of direction changes (forward $\leftrightarrow$ backward) in the trajectory, reflecting maneuver smoothness. These metrics jointly capture robustness, efficiency, and practicality of the planner in constrained parking scenarios.

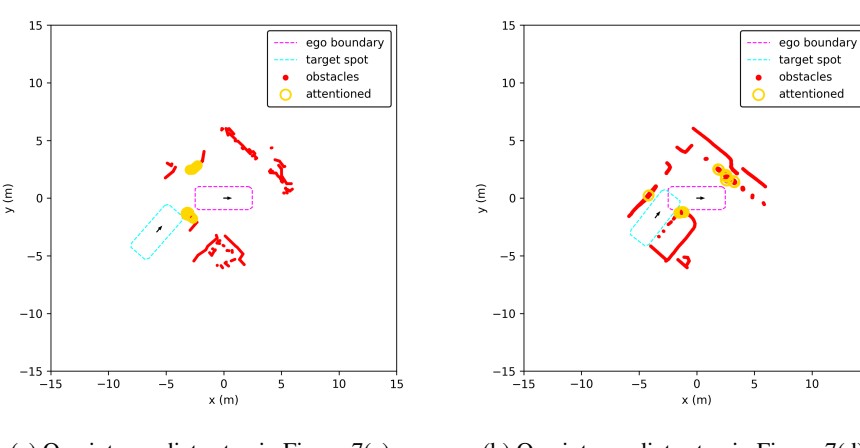

(a) One intermediate step in Figure 7(a).  (b) One intermediate step in Figure 7(d).

Figure 6: Examples of attention maps for one single decision-making step, respectively, in ego frame. We highlight the top 20 attention weights over the obstacle points.

## 4.2 RESULTS COMPARE

We compare our method against the Hybrid A* baseline on **ParkBench**. In addition to our full model, we also report a variant that uses PPO with curriculum learning but *without* action chunking to isolate the effect of chunking. Both learning methods start from the same logged ego poses as Hybrid A*. Results are summarized in Table 1. All evaluations are conducted on the same laptop with **CPU** Intel 12th Gen Core i5-1245U, **Python** 3.9, **PyTorch** 2.6.0, **SB3** 2.2.1. No GPU usage.

Our RL planner outperforms Hybrid A* across nearly all metrics, achieving higher success rates, substantially lower planning time, shorter paths, and comparable pivot counts, indicating smooth and

more efficient maneuvers in constrained settings. For clarity, Table 1 reports the classical baseline, PPO+Curriculum, and our full model (+Action Chunking). **We omit plain PPO and PPO+Action-Chunking in Table 1 because, without curriculum learning, both variants fail to acquire the parking behavior and achieve nearly zero success**. We also implemented standard SAC, DQN, DDPG, and other popular off-the-shelf RL algorithms (Andrychowicz et al., 2017). Without the proposed action-chunking wrapper, these methods achieve near-zero success rates. Therefore, we excluded them from the table for clarity.

### 4.3 QUALITATIVE RESULTS

Figure 7 shows some representative success cases from the parking scenarios in **ParkBench**. The examples demonstrate that our RL policy can generate human-like paths and is capable of conducting long-horizon planning. We also visualize the attention maps from the feature extractor module to verify that the model can correctly identifies the obstacles most critical for planning at a given time frame. The plots in Figure 6 show that, at the current ego pose, the agent appropriately attends to the relevant obstacles.

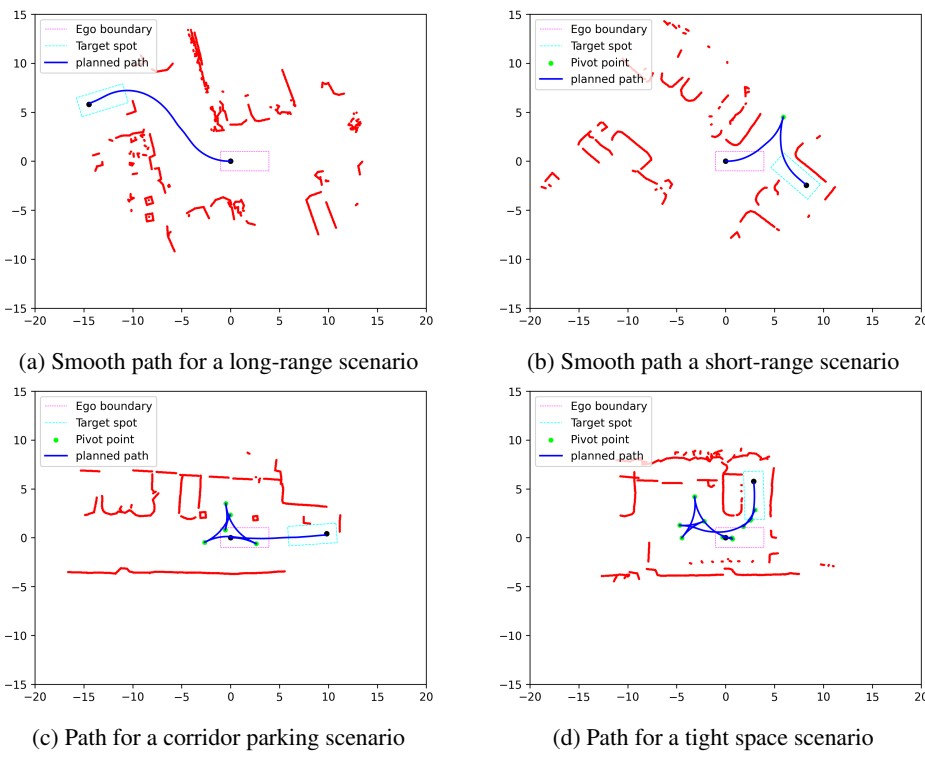

(a) Smooth path for a long-range scenario

(b) Smooth path a short-range scenario

(c) Path for a corridor parking scenario

(d) Path for a tight space scenario

Figure 7: Representative planned paths generated in different parking scenarios. Panels (a) and (b) show smooth paths from the ego vehicle's start pose (magenta) to the target pose (cyan), avoiding obstacles (red). Panels (c) and (d) illustrate highly constrained cases where the planner introduces multiple pivot points, resulting in non-optimal but collision-free paths.

### 4.4 REAL-VEHICLE DEPLOYMENT

Since our RL-based planner is designed with the goal of replacing the classical planner in the current pipeline, deployment on a real vehicle is straightforward. We first export the trained RL checkpoint to a compact C++ inference module (via ONNX) and integrate it into the onboard planning stack. At runtime, the RL planner receives the target pose from the user along with obstacle information from the perception system, and generates a collision-free reference path in the ego frame. This path is then tracked by a standard controller, which produces steering and velocity commands consistent with vehicle-dynamics and comfort constraints, enabling seamless execution of the learned policy

on the real vehicle. The deployment was conducted on our in-house test platform using the trained policy. The demonstration videos currently remain internal due to organizational policy, but we will be able to share more details in a future release.

### 4.5 LIMITATIONS

Through extensive evaluation across diverse rear-in parking scenarios to assess the generalization capability of the learned policy, we identify two main limitations.

**(1) Degraded performance in open/empty spaces.** While the planner performs well in tightly constrained environments (remains $90\%+$), its success rate drops in sparsely constrained scenes. We hypothesize the cause: empty-space scenarios were underrepresented during training. Future work include augmenting observations with free-space/clearance features and incorporating empty-space cases into the curriculum.

**(2) Manually specified curriculum.** The present eight-stage curriculum is hand-crafted for rear-in parking and does not transfer cleanly to other maneuvers (*e.g.*, parallel parking), limiting scalability and parallelization of training. Future work include exploring automatic curricula to broaden the task coverage.

## 5 CONCLUSION

In this paper, we presented an RL framework for path planning in constrained parking spaces. We introduced **ParkBench**, a benchmark tailored to diverse and realistic parking scenarios, and designed a training methodology that integrates a plug-in action chunking wrapper with curriculum learning. Our approach achieves state-of-the-art performance, outperforming a classical Hybrid A* baseline by a significant margin. We open-sourced all layouts, vehicle parameters, and our RL training methodology to encourage broader community adoption and improvement. Our goal is to provide a standardized reference framework that others can build upon, refine, and potentially surpass.

For future work, we plan to expand **ParkBench** with additional scenarios to cover a broader range of parking maneuvers, including head-in and parallel parking. We also want to improve the scalability of our RL training methodology by developing an automatic curriculum learning scheme.

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

APPENDIX

## A  USAGE OF LARGE LANGUAGE MODELS

During the preparation of this work, the authors used **ChatGPT-5**, a large language model, for grammar and language editing. However, all content was subsequently reviewed and revised by the authors for correctness, and the authors take full responsibility for the final manuscript.

## B  VEHICLE AND BICYCLE-MODEL PARAMETERS

In this work, we use the bicycle model to update the environment. The detailed parameters for the bicycle model are listed below (Table 2):

| Name | Symbol | Value | Unit / Notes |
|------|--------|-------|--------------|
| Wheelbase | $W_B$ | 3.0 | m |
| Vehicle width | $W$ | 2.0 | m |
| Vehicle length | $L$ | 4.95 | m |
| Rear overhang (rear center $\rightarrow$ bumper) | $L_B$ | 1.025 | m |
| Front overhang (rear center $\rightarrow$ front bumper) | $L_F$ | 3.925 | m |
| Max steering angle | $\delta_{\max}$ | $32°$ | deg |

Table 2: Physical and geometric parameters used by the kinematic bicycle model. The vehicle reference frame is the *rear-axle center* at the origin, $+x$ forward, $+y$ to the left.

Due to the tight space in the parking scenarios, we adopt a precise polygon footprint for collision check instead of a plain rectangle. The polygon is constructed by cropping each corner of the rectangle by a longitudinal offset $crop_l$ and a lateral offset $crop_w$. The resulting eight-vertex polygon is defined in the vehicle frame (rear-axle center at the origin, $+x$ forward, $+y$ left):

$$\texttt{polygon} = \begin{bmatrix} -L_B + crop_l & -W/2 \\ L_F - crop_l & -W/2 \\ L_F & -W/2 + crop_w \\ L_F & W/2 - crop_w \\ L_F - crop_l & W/2 \\ -L_B + crop_l & W/2 \\ -L_B & W/2 - crop_w \\ -L_B & -W/2 + crop_w \end{bmatrix},$$

where $crop_l = 0.3$ m and $crop_w = 0.2$ m.

## C  RL ALGORITHM SETTINGS

The following table (Table 3) summarizes the hyperparameter settings used in our RL training. The maximum episode length varies across curriculum stages. Specifically, we set it to $[100, 200, 400, 400, 800, 800, 800, 1000]$ for the 8 stages, respectively. All other parameters not listed are kept at their default values.

| Parameter | Value |
|---|---|
| Training batch size | 256 |
| Batch size per GPU | 1024 |
| Num. PPO epochs | 10 |
| Discount factor $\gamma$ | 1.0 |
| Max. episode length | depends on the training stage |
| Initial LR $\alpha^{(0)}$ | $3 \times 10^{-4}$ |
| LR schedule | Constant |
| Entropy coefficient | 0.001 |
| GPU usage during inference | Not used |

Table 3: RL algorithm settings and hyperparameters used during training.

## D  ENVIRONMENT ACTION SPACE

In our environment design, we use a discrete action space (shown in Table 4) parameterized by the steering increment $\Delta\delta$ and a signed speed $v$. In our environment, $\Delta\delta \in \{-8, 0, 8\}$ and $v \in \{-0.8, 0, 0.8\}$. The eight actions listed below are derived from statistics of motion primitives generated by our classical planner; this choice yields smooth curvature changes and reliable tracking under a pure-pursuit controller. We exclude the idle $(0,0)$ action to avoid no-operation steps.

Table 4: Discrete action space used in all RL experiments. Each action is a primitive $(\Delta\delta, v)$ with steering increment $\Delta\delta$ in degrees and longitudinal speed $v$ in m/s. Time step is $0.1s$.

| Index | $\Delta\delta$ [deg] | $v$ [m/s] | Description |
|---|---|---|---|
| 0 | $-8$ | $+0.8$ | Turn right, forward |
| 1 | $0$ | $+0.8$ | Straight, forward |
| 2 | $+8$ | $+0.8$ | Turn left, forward |
| 3 | $-8$ | $-0.8$ | Turn right, reverse |
| 4 | $0$ | $-0.8$ | Straight, reverse |
| 5 | $+8$ | $-0.8$ | Turn left, reverse |
| 6 | $-8$ | $0$ | Pre-steer right (no translation) |
| 7 | $+8$ | $0$ | Pre-steer left (no translation) |

## E  HYBRID A* HYPERPARAMETERS

We adopt the public available path planning repository as mentioned in section 2.1. To accelerate path searching, we exclude obstacle points located more than 25 meters from the ego's initial position. The bicycle model follows the same configuration as our simulation environment, with a wheelbase of 3.0 m, a width of 2.0 m, a length of 4.95 m, and a maximum steering angle of $32°$. Table 5 shows the ablation study we conducted on the hyperparameters of Hybrid A*. We report the best-performing configuration in the paper.

## F  RL ALGORITHM

Pseudocode: integrating curriculum learning and an action-chunking wrapper with the PPO algorithm. Shown in Algorithm 1.

| Heuristic | $\Delta x, y$ (m) | $\Delta\theta$ | Motion res. (m) | #Steer | Success (%) | Time (s) | Dist. (m) | Pivots |
|---|---|---|---|---|---|---|---|---|
| | 0.1 | 8° | 1.0 | 9 | 37.3 | 3.64 | 23.1 | 3.6 |
| | 0.32 | 8° | 1.0 | 9 | 41.2 | 0.41 | 22.8 | 4.1 |
| | 0.5 | 8° | 1.0 | 9 | 47.1 | 0.74 | 23.4 | 3.7 |
| Default values† | 0.5 | 8° | 0.5 | 9 | 45.1 | 0.64 | 24.3 | 3.7 |
| | 0.5 | 8° | 2.0 | 9 | 29.4 | 0.70 | 25.9 | 1.7 |
| | **0.5** | **5°** | **1.0** | **20** | **47.1** | **0.42** | **22.3** | 3.2 |
| | 1.0 | 5° | 1.0 | 20 | 41.2 | 0.96 | 23.8 | 1.7 |
| | 0.1 | 8° | 1.0 | 9 | 35.3 | 5.29 | 20.0 | 3.6 |
| | 0.32 | 8° | 1.0 | 9 | 41.2 | 0.53 | 19.8 | 4.1 |
| New values‡ | 0.5 | 8° | 1.0 | 9 | 43.1 | 0.69 | 19.3 | 3.9 |
| | 0.5 | 8° | 2.0 | 9 | 29.4 | 0.82 | 24.8 | 1.8 |
| | 0.5 | 8° | 0.5 | 9 | 43.1 | 0.5 | 19.9 | 3.6 |

Table 5: Ablation study on Hybrid A* hyperparameters.

*Notes.* †The default values correspond to the existing heuristics in the public repository. ‡The new values correspond to our new experiments. Specifically, we set the switch-back penalty cost to 2.0, the backward penalty cost to 1.3, the steering angle penalty cost to 0.2, the steering change penalty cost to 0.1, and the heuristic cost to 1.0.

---

**Algorithm 1** Training RL Planner with Action Chunking and Curriculum Learning

---

**Require:** Benchmark $\mathcal{B}$, simulator `Env`, policy $\pi_\theta$, chunk length $h$, curriculum scheduler $\mathcal{C}$, PPO optimizer, total_steps $N$

1: Initialize rollout buffer $\mathcal{D} \leftarrow \emptyset$, policy params $\theta$
2: **for** iteration $= 1, 2, \ldots$ **do**
3:     **Select curriculum level** $c \leftarrow \mathcal{C}(\text{iteration})$
4:     step $\leftarrow 0$
5:     **while** step $< N$ **do**
6:         **Sample scenario** $(g, O) \sim \mathcal{B}$         ▷ $g$: target pose, $O$: obstacle contours
7:         **Rollout init**: $p_0 \leftarrow$ ROLLOUTFROMTARGET$(g, O, c)$     ▷ $p_0$: ego initial pose
8:         **Reset env**: `Env`.RESET$(g, O, p_0)$
9:         **state**: $s_0 \leftarrow$ EgoCentric$(p_0, g_0, O_0)$   ▷ coordinate transform, normalization, range clip
10:        done $\leftarrow$ False
11:        **while** not done **do**
12:           **Chunked action**: $a_{t:t+h} \sim \pi_\theta(s_t)$        ▷ $a_{t:t+h}$ encodes $h$ primitive steps
13:           $R \leftarrow 0$
14:           **for** $k = 0$ to $h - 1$ **do**       ▷ Action chunk wrapper executes $h$ low-level steps
15:             $(p_{t+k+1}, g_{t+k+1}, O_{t+k+1}, r, \text{done}) \leftarrow$ `Env`.STEP$(\text{primitive}(a_{t+k}))$
16:             $R \leftarrow R + r; \; s \leftarrow$ EgoCentric$(p_{t+k+1}, g_{t+k+1}, O_{t+k+1})$
17:             **if** done **then break**
18:             **end if**
19:           **end for**
20:           Store transition $(s_t, a_t, R, s, \text{done})$ into $\mathcal{D}$
21:           $s_{t+1} \leftarrow s$
22:           **if** $\mathcal{D}$ is full **then**
23:             UPDATEPOLICYPPO$(\pi_\theta, \mathcal{D})$;
24:             $\mathcal{D} \leftarrow \emptyset$
25:           **end if**
26:         **end while**
27:     **end while**
28: **end for**

# G    LAYOUTS IN PARKBENCH

All 51 ParkBench parking-scenario layouts (17×3 grid):

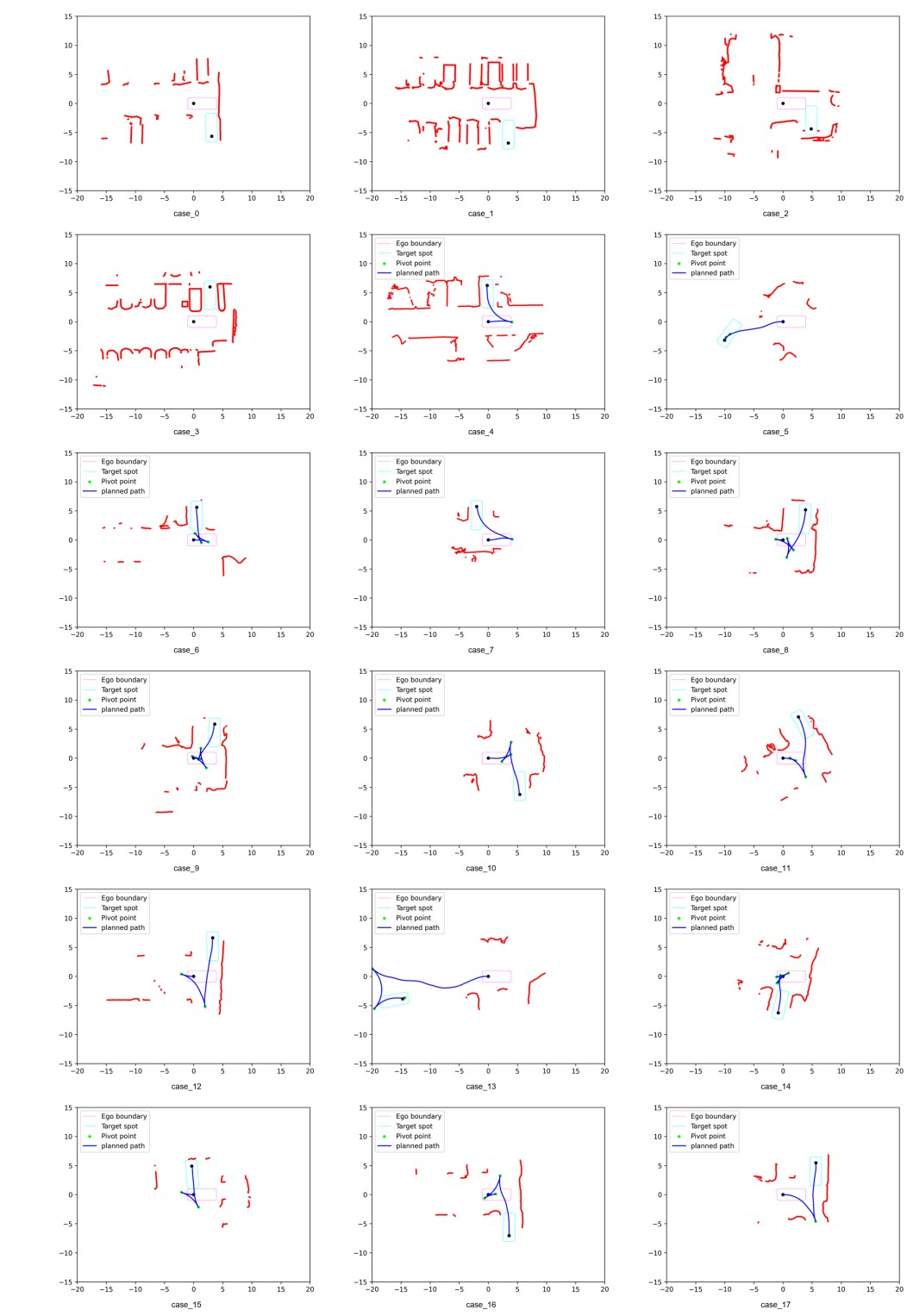

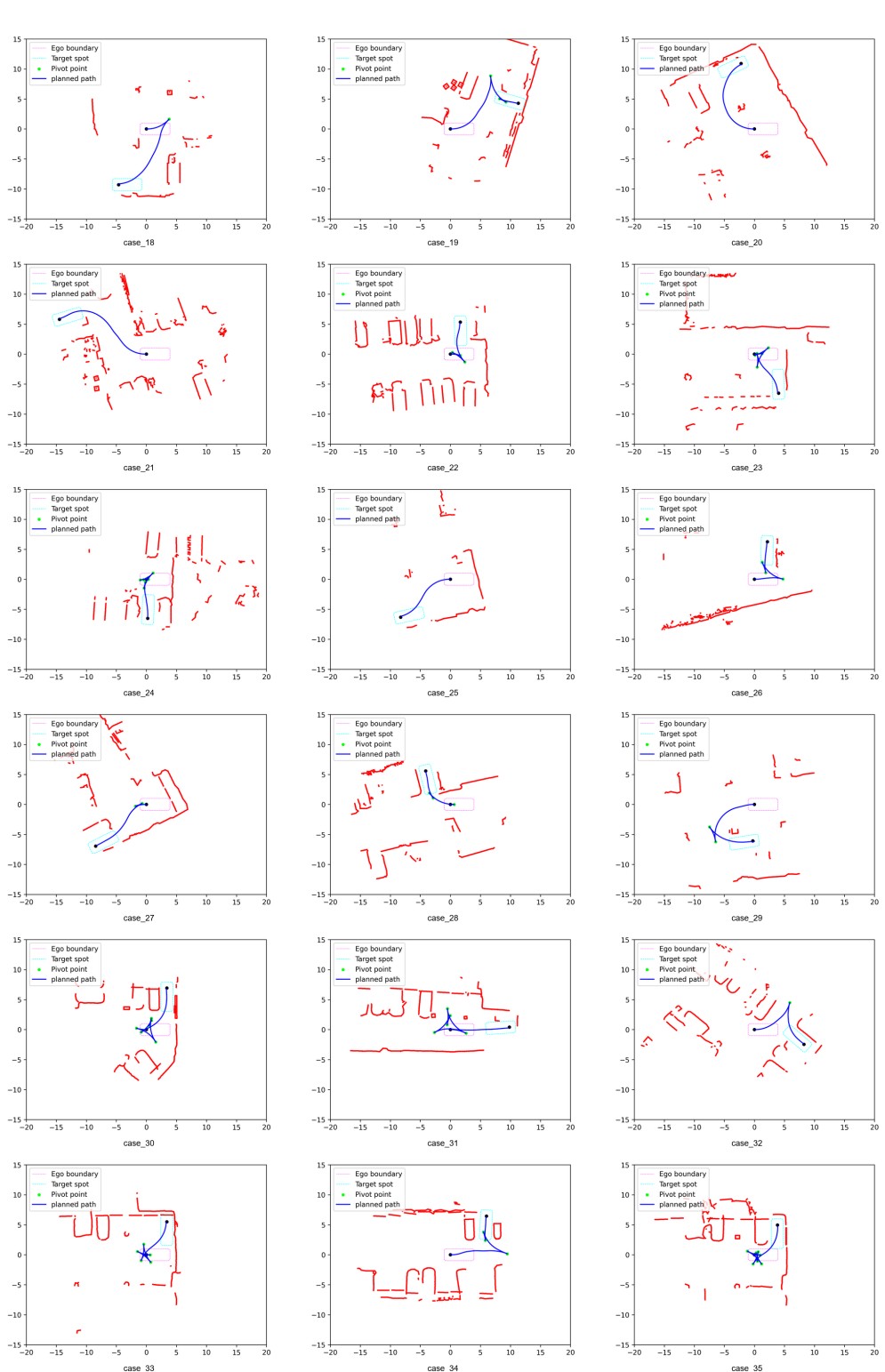

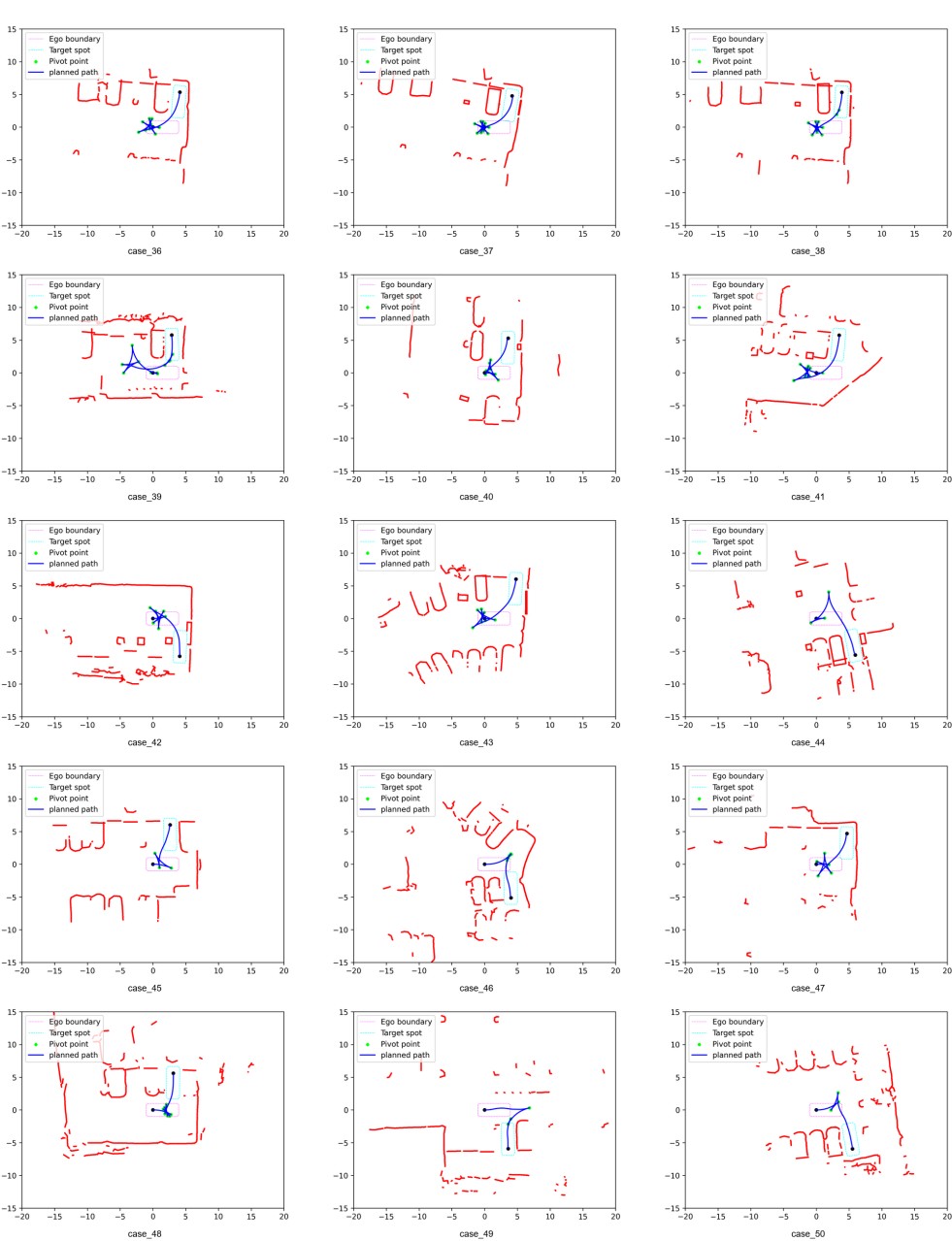

Figure 8: All 51 parking cases in the **ParkBench** benchmark, arranged in 17×3 grids across three pages. The magenta rectangle denotes the initial ego pose, while the cyan rectangle indicates the target pose. Obstacles are shown in red. The planned paths generated by our AI planner are illustrated in blue. The planner succeeds in 47 out of 51 scenarios, failing in 4 cases.

