# OpenReview forum: "Adapting Reinforcement Learning for Path Planning in Constrained Parking Scenarios"
_ICLR.cc/2026/Conference — Submitted to ICLR 2026_

### Official Review · Reviewer_JmMd · 2025-10-28

**Soundness:** 3
**Presentation:** 3
**Contribution:** 3
**Rating:** 4
**Confidence:** 3

**Summary:**

This paper proposes a reinforcement learning-based method for parking path planning using a bicycle kinematic model, demonstrating excellent performance in narrow and complex environments.
To address the challenges of long horizons, high-precision control, and sparse rewards, the approach introduces two key techniques: curriculum learning, which gradually progresses from simple to complex scenarios to improve training stability; and action chunking, which groups multiple primitive actions into macro-actions to balance exploration efficiency and maneuver precision. For fair comparison, the method uses the same inputs as Hybrid A: ego and target poses, and obstacle contours. A "rollout-back" mechanism is designed to generate feasible initial states, resolving infeasibility caused by sparse representation. Inputs are transformed into the ego frame, normalized, and constrained by a limited perception range to simulate real sensor limitations; a cross-attention mechanism enables the agent to focus on critical obstacles.
Experiments are conducted on the authors' self-constructed ParkBench benchmark, showing that the proposed method significantly outperforms Hybrid A in success rate, planning time, and travel distance. Ablation studies validate the effectiveness of curriculum learning and action chunking, while attention visualizations demonstrate strong environmental understanding. Despite degraded performance in open spaces and reliance on manually designed curricula, this work highlights the great potential of RL for generating human-like trajectories in complex parking tasks.

**Strengths:**

1.	The ParkBench is comprehensively designed and highly representative of real-world parking scenarios, making a significant contribution to the evaluation of parking trajectory planning methods. We look forward to its future expansion to cover even more diverse maneuvers.
2.	The paper effectively addresses a practical engineering challenge—initial pose infeasibility caused by sparse obstacle representation—through a well-designed roll-out mechanism that generates valid starting states, thereby improving training stability and realism.
3.	The integration of curriculum learning and action chunking significantly enhances both success rate and computational efficiency, demonstrating clear advantages over classical planning approaches in complex, constrained environments.

**Weaknesses:**

1.	The current curriculum is manually designed specifically for rear-in parking, which limits its transferability to other maneuvers such as parallel or angle parking. To enhance generalization and scalability, the authors are encouraged to explore automated curriculum learning methods that can adaptively generate training tasks across diverse scenarios.
2.	While the method performs exceptionally well in narrow and complex environments, its performance degrades in open or sparsely constrained spaces. Given that the current dataset primarily focuses on challenging narrow scenarios, it would be valuable to include more diverse environments—especially open layouts—in both training and evaluation, along with a detailed analysis of the model’s behavior across different scene types.
3.	Although the limited perception range simulates partial observability under real-world sensing constraints, all experiments are conducted in static environments. For a more realistic assessment, future work should evaluate the method’s robustness in dynamic settings involving moving obstacles such as pedestrians and vehicles, which are common in practical parking scenarios.

**Questions:**

1.	In Table 1, the line of"PPO (Ours) CL✓ Chunking✗" shows a significantly higher number of pivot points (53.4) compared to Hybrid A (3.2). Does this indicate that, in the absence of action chunking, the policy tends to generate excessive and unnecessary forward-backward transitions? It would be helpful to provide visualizations of such trajectories and analyze the underlying causes of this oscillatory behavior—whether it stems from poor temporal credit assignment, suboptimal exploration, or instability in policy learning.
2.	The 51 scenarios in ParkBench are extracted from real-world datasets. Could the authors provide more details on the distribution of these scenarios? Specifically, how many correspond to perpendicular parking, parallel parking, or other configurations? A breakdown of scene types would help assess the benchmark's diversity and representativeness and clarify the scope of the method’s evaluation.

---

> ### Author Response · Authors · 2025-11-21
> **Response to Reviewer JmMd**
>
> _W1: The current curriculum is manually designed specifically for rear-in parking, which limits its transferability to other maneuvers such as parallel or angle parking. To enhance generalization and scalability, the authors are encouraged to explore automated curriculum learning methods that can adaptively generate training tasks across diverse scenarios._
>
> **Response for W1:**
>
> - We appreciate the reviewer’s comment and encouragement. This limitation is already discussed in the paper, and we fully agree that automated curricula could further improve generalization. We are actively exploring automatic curriculum generation, but we view this as a separate line of future work. In this paper, our goal is to establish the foundation, i.e., constructing ParkBench and demonstrating the feasibility of an RL-based planner for the rear-in configuration. Our approach is easily transferable to parallel and front in parking, and we are currently exploring this.
>
> _W2: While the method performs exceptionally well in narrow and complex environments, its performance degrades in open or sparsely constrained spaces. Given that the current dataset primarily focuses on challenging narrow scenarios, it would be valuable to include more diverse environments—especially open layouts—in both training and evaluation, along with a detailed analysis of the model’s behavior across different scene types._
>
> **Response for W2:**
>
> - The performance degrade is observed in open space scenarios but overall, the average performance still remains over 90% in unseen scenarios (for rear-in tasks). For the ParkBench dataset, we will soon extend it with additional parking types.
>
>
> _W3: Although the limited perception range simulates partial observability under real-world sensing constraints, all experiments are conducted in static environments. For a more realistic assessment, future work should evaluate the method’s robustness in dynamic settings involving moving obstacles such as pedestrians and vehicles, which are common in practical parking scenarios._
>
> **Response for W3:**
> + In our current autonomous parking system, dynamic obstacles (e.g., pedestrians or slowly moving vehicles) are handled by a separate safety module that performs real-time collision checking and can override planned motions when necessary. Considering our parking maneuver operates at low speeds, this safety layer is typically sufficient to ensure reliability in dynamic settings. It is worth mentioning that our RL-based planner already incorporates an attention mechanism that prioritizes obstacles along the predicted path (as illustrated in Figure 6), which we expect to help the policy adapt to moderate dynamic changes. However, we agree that a full evaluation in dynamic environments would provide a more realistic assessment. This is an important direction for future work, and we plan to investigate it as we continue extending the system.

---

> > ### Comment · Reviewer_JmMd · 2025-11-26
> >
> > Based on the rebuttal, the authors do not present a positive response to the comments. I change my rating to reject.

---

> > > ### Author Response · Authors · 2025-11-26
> > >
> > > Could the reviewer clarify what additional “positive response” was expected?
> > >
> > > We addressed all stated weaknesses and made concrete revisions (please refer to the revision of the paper, changes are highlighted in red), including adding two new visualizations in the Supplementary Material (since images cannot be pasted in the rebuttal). Given these updates, it is unclear which concerns the reviewer believes remain unresolved.

---

> ### Author Response · Authors · 2025-11-21
>
> ## For the Questions part:
>
> _Q1: In Table 1, the line of "PPO (Ours) CL✓ Chunking✗" shows a significantly higher number of pivot points (53.4) compared to Hybrid A (3.2). Does this indicate that, in the absence of action chunking, the policy tends to generate excessive and unnecessary forward-backward transitions? It would be helpful to provide visualizations of such trajectories and analyze the underlying causes of this oscillatory behavior—whether it stems from poor temporal credit assignment, suboptimal exploration, or instability in policy learning._
>
> **Response for Q1:**
>
> - Thanks for highlighting the need to analyze the underlying cause of the large number of pivot points. The higher pivot count in the checkpoint trained with curriculum learning but without the action-chunking wrapper stems from oscillatory behavior during planning. Without chunking, the policy frequently executes very short forward–reverse motions, producing many micro-maneuvers and therefore numerous geometric pivot points. These small movements can also trap the agent in locally optimal but globally inefficient correction loops, where making another short gear switch yields an immediate improvement in distance or orientation error, even though it increases long-horizon trajectory complexity. In addition, the reward term encouraging path smoothness is relatively weak compared to the strong terminal pose reward, so the policy receives little incentive to avoid these high-frequency adjustments. As shown in the following figure, two similar dead-end parking scenarios exhibit this behavior: the agent repeatedly makes short back-and-forth corrections, which amplifies pivot-point counts and leads to jagged trajectories. Introducing action chunking mitigates these issues by forcing longer, coherent motion segments, thereby reducing local oscillations, avoiding local optimal traps, and producing smoother, tracker-friendly paths.
>
> ### Note: It seems Openreview does not support images, we uploaded the figures to Supplementary Material instead (with the name: Figure 1a and Figure 1b).
>
> _Q2: The 51 scenarios in ParkBench are extracted from real-world datasets. Could the authors provide more details on the distribution of these scenarios? Specifically, how many correspond to perpendicular parking, parallel parking, or other configurations? A breakdown of scene types would help assess the benchmark's diversity and representativeness and clarify the scope of the method’s evaluation._
>
> **Response for Q2:**
>
> - Thank you for raising this important question. The 51 scenarios in ParkBench can be broadly categorized into two types: (1) corridor-style parking (\~24%) and (2) dead-end parking (\~76%). All scenarios correspond to rear-in perpendicular parking configurations. Figure 8 of this paper (Appendix G) provides visualizations of all 51 layouts along with the corresponding planned paths. For the reviewer’s convenience, we also include a visualization in which all target spots and obstacles from the 51 scenarios are projected into the ego coordinate system and overlaid into a single composite image. This aggregated plot is shown below.
>
> ### Note: It seems Openreview does not support images, we uploaded the figures to Supplementary Material instead (with the name: Figure 2).
>
> - We agree that the diversity and representativeness of a benchmark are crucial for developing planners that generalize across different parking situations. At this stage, our RL-based planner is primarily trained for rear-in tasks. However, we are actively expanding ParkBench to include additional parking types, such as head-in and parallel parking, to further broaden the benchmark’s coverage and support more comprehensive planner evaluations.

---

### Official Review · Reviewer_683i · 2025-10-29

**Soundness:** 2
**Presentation:** 2
**Contribution:** 1
**Rating:** 2
**Confidence:** 5

**Summary:**

This paper employs a RL–based approach to replace the classical planner in path planning tasks. It empirically demonstrates the effectiveness of the proposed method in parking scenarios.

**Strengths:**

1. The paper shows the effectiveness of using the PPO algorithm as a planner in parking scenarios.

2. It introduces a new benchmark, ParkBench, to facilitate research on path planning in parking environments.

**Weaknesses:**

1. In the introduction, the authors claim that RL-based methods, as representatives of closed-loop approaches, remain underexplored in path planning. However, RL-based planners have been extensively studied in various path planning tasks that consider practical constraints across diverse real-world applications, including transportation, warehousing, and surgical robotics.

2. Numerous studies have focused on developing RL-based planners in related domains. Although this paper centers on parking scenarios, this domain is closely connected to transportation and autonomous driving. The related work section does not sufficiently review these prior studies.

3. In the empirical evaluation, the proposed method is compared only with one classical heuristic baseline. The authors do not include adequate baseline methods, especially those that also utilize RL techniques. As a result, the experimental results do not convincingly demonstrate the contribution of this work.

4. The classical heuristic method used as the baseline has shown strong performance and generalization across other scenarios. To further validate the proposed method’s effectiveness and generalization, it would be beneficial to evaluate it on additional tasks.

5. In the experimental section, the authors integrate only the PPO algorithm into the parking planner. A comparison with other standard RL algorithms would provide a more comprehensive understanding of the framework’s robustness and performance.

6. The contribution of this paper is limited. It primarily applies an existing RL algorithm to a parking task, which is not a novel direction for the community, although the development of the ParkBench benchmark and consideration of practical constraints are appreciated.

**Questions:**

1. Since this work demonstrates the practicality of an RL-based planner trained on ParkBench, is it possible to deploy this method in a real vehicle, similar to how the A* algorithm has been applied?

---

> ### Author Response · Authors · 2025-11-21
> **Response to Reviewer 683i**
>
> _W1: In the introduction, the authors claim that RL-based methods, as representatives of closed-loop approaches, remain underexplored in path planning. However, RL-based planners have been extensively studied in various path planning tasks that consider practical constraints across diverse real-world applications, including transportation, warehousing, and surgical robotics._
>
> **Response for W1:**
>
> - Thanks for pointing this out. We have added one extra paragraph in section 2.2 to include the missing literature on the RL-based planner. We also modified our statement in the introduction (highlighted in red) to make it more accurate. In particular, we emphasize that RL-based planners have not been fully explored in the context of autonomous driving.
>
>
>
> _W2: Numerous studies have focused on developing RL-based planners in related domains. Although this paper centers on parking scenarios, this domain is closely connected to transportation and autonomous driving. The related work section does not sufficiently review these prior studies._
>
> **Response for W2:**
>
> - The weakness is closely related to W1, and we have added one paragraph in the related work section to address the insufficient review on prior works (highlighted in red).
>
> _W3: In the empirical evaluation, the proposed method is compared only with one classical heuristic baseline. The authors do not include adequate baseline methods, especially those that also utilize RL techniques. As a result, the experimental results do not convincingly demonstrate the contribution of this work._
>
> **Response for W3:**
>
> - Hybrid A* is the most widely adopted baseline in industry due to its strong performance and ability to generate kinematically feasible and tracker-friendly trajectories. For learning-based baselines, we implemented standard PPO, SAC, DQN, DDPG, and other popular off-the-shelf RL algorithms. Without the proposed action-chunking wrapper, these methods achieve near-zero success rates, demonstrating that naively applying generic RL algorithms is insufficient for this highly constrained parking task. To better demonstrate the contribution of our work, we have added descriptions of the failure cases of standard off-the-shelf RL algorithms in Section 4.2, with the changes highlighted in red.
>
>
> _W4: The classical heuristic method used as the baseline has shown strong performance and generalization across other scenarios. To further validate the proposed method’s effectiveness and generalization, it would be beneficial to evaluate it on additional tasks._
>
> **Response for W4:**
>
> - We agree with the reviewer’s suggestion. As noted in our limitations section, evaluating the planner on additional tasks beyond rear-in parking would further strengthen the generalization claims, and we plan to explore these extensions in future work.
>
>
> _W5: In the experimental section, the authors integrate only the PPO algorithm into the parking planner. A comparison with other standard RL algorithms would provide a more comprehensive understanding of the framework’s robustness and performance._
>
> **Response for W5:**
>
> - This point overlaps with W3. We evaluated multiple standard RL algorithms (PPO, SAC, DQN, DDPG, etc.), but without our action-chunking wrapper, their performance remained close to zero success rate. These results indicate that the proposed framework, rather than the choice of RL backbone, is the primary factor enabling reliable planning in this constrained setting. To address the weakness, we have added descriptions of the failure cases of standard off-the-shelf RL algorithms in Section 4.2, with the changes highlighted in red.
>
> _W6: The contribution of this paper is limited. It primarily applies an existing RL algorithm to a parking task, which is not a novel direction for the community, although the development of the ParkBench benchmark and consideration of practical constraints are appreciated._
>
> **Response for W6:**
>
> - We appreciate the reviewer’s recognition on our benchmark. However, we disagree with the reviewer’s conclusion that “the contribution of this paper is limited”. In fact, to solve the planner task, we began with a vanilla RL algorithm---what the reviewer describes as an existing RL algorithm---but the results were far from adequate, as shown in our paper. The basic RL setup fails entirely. We therefore introduced curriculum learning, which helped but was still insufficient. Only after incorporating an action-chunking approach to mitigate oscillatory behavior did we reach stable and reasonable performance.

---

> > ### Author Response · Authors · 2025-11-21
> >
> > ## For the Questions part:
> >
> > _Q1: Since this work demonstrates the practicality of an RL-based planner trained on ParkBench, is it possible to deploy this method in a real vehicle, similar to how the A* algorithm has been applied?_
> >
> > **Response for Q1:**
> >
> > Yes. We have successfully deployed our RL-based planner on a real vehicle.  The deployment was conducted on our in-house test platform using the trained policy, without any additional task-specific tuning. The demonstration videos currently remain internal due to organizational policy, but we will be able to share more details in a future release. To highlight our contribution on the real world deployments, we have added one extra paragraph to describe the in-vehicle tests in section 4.4 (highlighted in red).

---

### Official Review · Reviewer_efQf · 2025-10-31

**Soundness:** 2
**Presentation:** 1
**Contribution:** 1
**Rating:** 2
**Confidence:** 4

**Summary:**

The paper presents a new reinforcement learning based path planning system for parking problem in cluttered environments. The paper also introduces ParkBench, a new parking benchmark using bicycle model dynamic for simulating the environment with a mobile robot. The method demonstrates more than 96% success and 52% efficiency improvement compared to classical path planner.

**Strengths:**

* Strong empirical results: Achieves 92.2% success on ParkBench vs. 47.1% for Hybrid A*, and 2× improvement in time efficiency.
* Benchmark contribution: ParkBench fills a benchmark gap in parking evaluation, providing 51 realistic layouts for reproducibility and comparison.

**Weaknesses:**

* The RL system presented in this paper is fairly straight forward. The component includes a handcrafted curriculum for initial configuration and  motion primitive (action chunking). These components are well-established in the literature and the author does not demonstrate sufficient effort in integrating these components as a whole system.

* There exists a lot of RL-based motion planning for mobile robot, many of them are trained in high-fidelity simulator, such as Gazebo and IsaacLab. The author does not fully address these prior work to demonstrate the novelty of the system.
Xu, Zifan, et al. "Benchmarking reinforcement learning techniques for autonomous navigation." arXiv preprint arXiv:2210.04839 (2022).
Akmandor, Neşet Ünver, et al. "Deep reinforcement learning based robot navigation in dynamic environments using occupancy values of motion primitives." 2022 IEEE/RSJ international conference on intelligent robots and systems (IROS). IEEE, 2022.

* Writing tone: Some claims (e.g., “potentially eliminates need for localization and tracking”) are speculative and should be framed more cautiously. The use of language is not precise and formal for a research paper.

**Questions:**

Please see the weakness part

---

> ### Author Response · Authors · 2025-11-21
> **Response to Reviewer efQf**
>
> We thank the reviewer for recognizing the contributions of our work. Parts of our paper may not have been sufficiently clear, which has led to some misunderstandings by the reviewer. We will revise the writing to improve clarity and structure. Below, we address the reviewer’s concerns point-by-point and clarify the aspects that appear to have been misinterpreted.
>
> - *W1: The RL system presented in this paper is fairly straight forward. The component includes a handcrafted curriculum for initial configuration and motion primitive (action chunking). These components are well-established in the literature and the author does not demonstrate sufficient effort in integrating these components as a whole system.*
>
>
> **Response for W1:**
>
> Despite the training pipeline has ingredients that reuse methodologies from literature (PPO, and curriculum learning), no one had studied extensively how to make them work for a real-time, in-vehicle-deployable, real-perception robust parking use case, especially in tight spots requiring many maneuvers. In particular, our feature extractor and policy structure are intentionally lightweight to ensure feasibility for real-time, on-device deployment, which is a key requirement of practical parking systems. Our curriculum learning also involves generating start positions automatically based on the rollout generation method as shown in the Figure 2 of the paper.
>
> The reviewer’s summary does not fully capture the core contribution of our system. In particular, our action-chunking wrapper is **not equivalent** to motion primitives as commonly defined in robotics literature. Classical motion primitives roll out a single control command for a fixed duration. In contrast, our action-chunking wrapper modifies the agent’s action space itself: the policy outputs a sequence of low-level actions in a single decision step. This fundamentally changes the temporal abstraction of RL and is especially beneficial in precision-maneuvering tasks such as the parking task we are investigating, where the agent must reason over short, highly constrained motions. Moreover, this wrapper is designed in a generic and modular fashion, similar in spirit to standard Gymnasium wrappers such as *FrameStack*. It can be readily integrated with any off-the-shelf RL algorithm without special tuning. This design provides a practical and reusable tool for RL-based planners. To demonstrate the effectiveness of our combination of (1) curriculum learning, (2) action chunking wrapper, and (3) ParkBench dataset, we conduct different training experiments. As shown in Table 1, removing either component leads to significant performance degradation, confirming that the integration of these ideas is crucial for achieving high success rates in constrained-space parking. We also evaluated the generalizability of our learned model in unseen scenarios and observed consistent performance.
>
> - _W2: There exists a lot of RL-based motion planning for mobile robot, many of them are trained in high-fidelity simulator, such as Gazebo and IsaacLab. The author does not fully address these prior work to demonstrate the novelty of the system. Xu, Zifan, et al. \"Benchmarking reinforcement learning techniques for autonomous navigation.\" arXiv preprint arXiv:2210.04839 (2022). Akmandor, Neşet Ünver, et al. \"Deep reinforcement learning based robot navigation in dynamic environments using occupancy values of motion primitives.\" 2022 IEEE/RSJ international conference on intelligent robots and systems (IROS). IEEE, 2022._
>
> **Response for W2:**
>
> We thank the reviewer for pointing out the missing information in the Related Works section and for suggesting two representative references. To better highlight the novelty of our RL system, we will add an additional paragraph to Section 2.2. For the reviewer’s convenience, we include the revised paragraph below:
>
> >Deep reinforcement learning (DRL) has been widely explored in mobile robot navigation [1]. However, these works are not directly applicable to the parking task we study in this work. Most navigation methods [2,3,4,5] assume differential-drive robots with highly flexible motion capabilities, whereas parking requires vehicle modeling governed by nonholonomic constraints such as the bicycle or the Ackermann-steering models. These kinematic models restrict maneuverability. A related study uses an RC-car platform and combines model-free and model-based RL for indoor navigation [6]. Despite these efforts, navigation goals are typically treated as waypoints without enforcing precise final orientation, while parking requires exact terminal conditions. To the best of our knowledge, few learning-based methods jointly consider these constraints, motivating the development of our RL-based parking planner.

---

> > ### Author Response · Authors · 2025-11-21
> >
> > Reference:
> >
> > [1] Zhu, Kai, and Tao Zhang. "Deep reinforcement learning based mobile robot navigation: A review." Tsinghua Science and Technology 26.5 (2021): 674-691.
> >
> > [2] C. Pérez-D’Arpino, C. Liu, P. Goebel, R. Martín-Martín and S. Savarese, "Robot Navigation in Constrained Pedestrian Environments using Reinforcement Learning," 2021 IEEE International Conference on Robotics and Automation (ICRA), Xi'an, China, 2021, pp. 1140-1146
> >
> > [3] X. Ruan, D. Ren, X. Zhu and J. Huang, "Mobile Robot Navigation based on Deep Reinforcement Learning," 2019 Chinese Control And Decision Conference (CCDC), Nanchang, China, 2019, pp. 6174-6178
> >
> > [4] Xu, Zifan, et al. "Benchmarking reinforcement learning techniques for autonomous navigation." arXiv preprint arXiv:2210.04839 (2022).
> >
> > [5] G. Kahn, A. Villaflor, B. Ding, P. Abbeel and S. Levine, "Self-Supervised Deep Reinforcement Learning with Generalized Computation Graphs for Robot Navigation," 2018 IEEE International Conference on Robotics and Automation (ICRA), Brisbane, QLD, Australia, 2018, pp. 5129-5136
> >
> > [6] Akmandor, Neşet Ünver, et al. "Deep reinforcement learning based robot navigation in dynamic environments using occupancy values of motion primitives." 2022 IEEE/RSJ international conference on intelligent robots and systems (IROS). IEEE, 2022.

---

> > > ### Author Response · Authors · 2025-11-21
> > >
> > > In addition, we note that high-fidelity simulators such as Gazebo or IsaacLab are often impractical for real-world parking applications, as modeling of tight parking geometries, diverse curb shapes, and occlusion patterns requires substantial manual effort and customization, and these simulators also demand significant CPU resources. To complement this gap, our ParkBench dataset provides realistic parking layouts collected from real-world environments. This dataset enables systematic evaluation of RL-based planners under genuine geometric constraints, offering a practical, lightweight alternative where high-fidelity simulation is unavailable.
> > >
> > >
> > > - _W3: Writing tone: Some claims (e.g., \"potentially eliminates need for localization and tracking\") are speculative and should be framed more cautiously. The use of language is not precise and formal for a research paper._
> > >
> > > **Response for W3:**
> > >
> > > The task we aim to solve in this paper is to replace the current classical planner module with a more robust alternative, i.e., one that can tolerate perception errors, and limited sensor coverage (e.g., occlusions), while remaining computationally efficient for onboard devices and friendly to the downstream path tracker. In the current implementation, our RL-based planner still relies on other components such as localization and tracking. Although the RL-based planner is capable of directly producing control commands and thus has the potential to reduce the need for additional modules, this capability is not yet utilized in our deployment. In the original writing, we intended to highlight this future potential while maintaining an accurate description of the present system. To avoid a speculative tone, we rephrase the sentence and removed the term \"potentially\" in the revision and carefully reviewed the rest of the paper to ensure that the wording remains precise and appropriate.

---

### Official Review · Reviewer_cNCZ · 2025-11-01

**Soundness:** 3
**Presentation:** 3
**Contribution:** 3
**Rating:** 6
**Confidence:** 3

**Summary:**

The authors present an approach for real-time path planning in tight spaces with Deep Reinforcement Learning. The paper shows how the approach outperforms an A* based baseline by a large margin. In addition to the paper they provide a benchmark for parking in tight spaces based on 2d coordinates, like lidar points showing the surroundings.

**Strengths:**

The performance compared to the baseline is quite impressive.

The paper also contributes a benchmark.

Including a bicycle model introduces an easily tunable component, if more complex vehicle models are needed, and that should ensure feasible trajectories.

The benchmark provides very simple lidar scans which may seem like a disadvantage but is an easily transferable representation that should be easy to provide in many contexts. It is also very easy to simulate as the paper shows.

The approach, and architecture, seems surprisingly simple.

**Weaknesses:**

While having impressive results, they are on a novel self-created benchmark which did not give other authors opportunity to optimize on it. Therefore, results have to be taken with a grain of salt and if the benchmark is accepted in the domain time must tell how these results hold up.

It seems more direction changes are needed, compared to a simple A* algorithm. While other metrics indicate better performance it would be interesting how this scales, i.e. how many more pivot points, which should come with a more complex trajectory, are needed for what performance boost?

The approach only compares to Hybrid A* and no other approach. It would have been possible to compare with other non-deep learning methods like a Reeds-Shepp Curve planner or Dijkstra.

Finally, this is a very good solution for this task but I am not completely convinced this is the right venue for it. Maybe IROS, IV or ITSC would have been more suitable. I leave that decision to the AC.

**Questions:**

Given the many penalties which are not primary goals, such as the goal achievement and collision, what are local minimas that were observed during training. In other words, were there "cheating" behaviors where the model optimized not getting penalized for being idle e.g. by moving very slow.

Please note that concurrent work on the horizon should be compared with this work in the future. E.g. "RAFT: Regularized Adversarial Fine-Tuning to Enhance Deep Reinforcement Learning for Self-Parking
, Pighetti et al.". This was published in August 2025 so it is irrelevant for judging this work.

---

> ### Author Response · Authors · 2025-11-21
> **Response to Reviewer cNCZ**
>
> We thank the reviewer for the careful reading and clear summary of our contributions. Regarding the comment in Strengths that ''*the approach and architecture seem surprisingly simple*,'' this simplicity is intentional: our architecture is designed for easy deployment on onboard devices. While we use a standard PPO backbone, we introduce an action-chunking wrapper that enables stable control over long-horizon, fine-grained maneuvers. This component is lightweight, broadly applicable, and can be readily integrated into other RL algorithms facing similar precision-movement challenges.
>
> ## For the Weaknesses part:
>
> - *W1: While having impressive results, they are on a novel self-created benchmark which did not give other authors opportunity to optimize on it. Therefore, results have to be taken with a grain of salt and if the benchmark is accepted in the domain time must tell how these results hold up.*
>
> **Response for W1:**
>
> We thank the reviewer for pointing out this and we would like to detail here why the release of the **ParkBench** dataset can address the concern. ParkBench was introduced because (1) no existing benchmark captures challenging parking scenarios with limited, occluded LiDAR/freespace observations, and (2) real-world experiments show that such constrained layouts are particularly challenging for classical planners. To assess generality of our trained model, we have evaluated it on cases outside the benchmark and observed approximately 90% success. This shows that ParkBench captures a comprehensive variety of parking situations. We open-sourced all layouts, vehicle parameters, and our RL training methodology to encourage broader community adoption and improvement. Our goal is to provide a standardized reference framework that others can build upon, refine, and potentially surpass.
>
>
> - *W2: It seems more direction changes are needed, compared to a simple A\* algorithm. While other metrics indicate better performance it would be interesting how this scales, i.e. how many more pivot points, which should come with a more complex trajectory, are needed for what performance boost?*
>
> **Response for W2:**
>
> We appreciate the reviewer’s thought on the relation between performance and the number of pivot points. In deployment, we favor less pivot points for the ease of path tracking. Our current planner still occasionally exhibits a greedy local-adjustment behavior, that is to adjust its pose via short movements within a small region (shown in Figure 8 of our paper, cases 35-38). Reducing such unnecessary pivots while preserving overall performance is an important direction that we are actively pursuing.  However, it is worth mentioning that our current solution already reaches comparable pivot counts compared to the hybrid A* planner as shown in Table 1 of the paper.
>
>
> - *W3: The approach only compares to Hybrid A\* and no other approach. It would have been possible to compare with other non-deep learning methods like a Reeds-Shepp Curve planner or Dijkstra.*
>
> **Response for W3:**
>
> We appreciate the reviewer’s suggestion to include additional classical planners such as Dijkstra, RRT, Reeds-Shepp curve, etc. However, based on our observation and the practical need (kinematically feasible trajectories suitable for tracking), Hybrid A* is the most popular baseline among the classical planners [1]. For learning-based baselines, we experimented with standard PPO, SAC, DQN, DDPG, and other off-the-shelf RL algorithms. Without the proposed action-chunking wrapper, their success rates were close to zero, indicating that naively applying generic RL algorithms is insufficient for this constrained parking task. As part of our future work, we plan to include modern ML-based planners (e.g., diffusion models). We also encourage researchers to work with our benchmark to speed up the development of a robust path planner for real-world deployment.
>
> Reference:
>
> [1] Zhang, P., Zhou, S., Hu, J. et al. Automatic parking trajectory planning in narrow spaces based on Hybrid A* and NMPC. Sci Rep 15, 1384 (2025). https://doi.org/10.1038/s41598-025-85541-x
>
>
> - *W4: Finally, this is a very good solution for this task but I am not completely convinced this is the right venue for it. Maybe IROS, IV or ITSC would have been more suitable. I leave that decision to the AC.*
>
> **Response for W4:**
>
> We appreciate the reviewer’s perspective on venue selection. Our submission aligns well with the ICLR domain track on applications to robotics, autonomy, and planning. Presenting this work at ICLR can help bridge robotics-oriented planning problems with advances in representation learning and reinforcement learning, and we believe it will encourage broader AI community engagement in developing learning-based planners for real-world parking and related tasks.

---

> ### Author Response · Authors · 2025-11-21
>
> ## For the Questions part:
>
> - _Q1: Given the many penalties which are not primary goals, such as the goal achievement and collision, what are local minimas that were observed during training. In other words, were there \"cheating\" behaviors where the model optimized not getting penalized for being idle e.g. by moving very slow._
>
> **Response for Q1:**
>
> This is an excellent question and a key challenge in training RL agents. The reward design presented in the paper reflects the best configuration we found after extensive experimentation. We did observe reward-hacking behaviors. For example, when increasing the collision penalty, the agent sometimes learned to remain idle in tight spaces to avoid penalty, as the reviewer noted. This is undesirable for parking, so the reward function must be carefully balanced to discourage both collisions and overly conservative behavior.
>
>
> - _Q2: Please note that concurrent work on the horizon should be compared with this work in the future. E.g. "RAFT: Regularized Adversarial Fine-Tuning to Enhance Deep Reinforcement Learning for Self-Parking , Pighetti et al.". This was published in August 2025 so it is irrelevant for judging this work._
>
> **Response for Q2:**
>
> We thank the reviewer for pointing out the concurrent work and appreciate the understanding regarding the timing of our writing. The referenced paper also studies parking, but its problem setting remains fully simulated in CARLA and covers relatively simple layouts. Our benchmark, however, emphasizes constrained, real-world–derived layouts and partial observations. We have carefully reviewed this and other related work and incorporated the appropriate citations and discussion in the revision (refer section 2.2, highlighted in red).

---

> > ### Comment · Reviewer_cNCZ · 2025-11-27
> >
> > Thank you for the detailed responses. Given negative points other reviewers mentioned which I overlooked I will leave my rating. I think the contribution could be still better suited for ITSC or IV which are strong venues for control for autonomous driving. The focus on centimeter-accurate terminal conditions seems not well motivated and more made to increase the relevance of this work. Most parking would be probably sufficient using waypoints and only some special hard parking cases would motivate more precision.

---

> > > ### Author Response · Authors · 2025-11-27
> > > **Thanks for the feedback on our revision**
> > >
> > > We appreciate the reviewer's careful reading of our paper and the timely follow-up comments on our revisions. While many parking scenarios can indeed be handled by classical planners given reliable perception input, there remain challenging cases where classical methods struggle, particularly under occlusions, in highly constrained geometries where perception is less robust, or strict real-time response requirements. These practical limitations motivate our focus on learning-based approaches.
> > >
> > > Since there is no strong baseline benchmark for AI-driven planners in this domain, we have open-sourced our challenging parking-layout environments. Our goal is to provide a standardized benchmark that encourages the community to explore AI-based solutions for real-world tasks and helps fill the missing pieces in the autonomous driving stack.
> > >
> > > Again, we sincerely thank the reviewer for the constructive feedback and supportive remarks. The comments have been very helpful in improving our paper.

---

### Comment · Area_Chair_G6Cr · 2025-11-24

Dear Reviewers,

The authors have responded to your reviews. Please review and respond to their comments who have not yet done so.

Best, Your AC

---

> ### Author Response · Authors · 2025-11-26
> **Summary of Reviewer Comments and Corresponding Rebuttal Actions**
>
> We sincerely thank all reviewers for their careful evaluations and constructive feedback. We are encouraged that reviewers consistently highlighted the strengths of our work, including (1) **our strong empirical performance (reviewers cNCZ, efQf, 683i, JmMd)** and (2) **the usefulness of the ParkBench benchmark (reviewers cNCZ, efQf, 683i, JmMd)**. Reviewers also noted that the system is simple yet effective (reviewer cNCZ), demonstrates large performance gains over classical planners (reviewer efQf), and tackles a practical and challenging real-world task (reviewer JmMd). We appreciate the reviewers for their careful reading and positive assessments.
>
> Below, we summarize the **three key weaknesses** mentioned by the reviewers and address the concerns and misunderstandings accordingly.
>
> - ### **Insufficient Baseline Coverage and Ablation Analysis** (Reviewers cNCZ, 683i, and JmMd)
>
> We clarify that:
>
> Based on our observation and the practical need (kinematically feasible trajectories suitable for tracking), Hybrid A* is the most popular baseline among the classical planners [1]. For learning-based baselines, we experimented with standard PPO, SAC, DQN, DDPG, and other off-the-shelf RL algorithms. Without the proposed action-chunking wrapper, their success rates were close to zero, indicating that naively applying generic RL algorithms is insufficient for this constrained parking task.
>
> To include the efforts of our ablation studies, **we have added the descriptions of the failure cases of standard off-the-shelf RL algorithms in Section 4.2 with the changes highlighted in red in the Revision**. These clarifications and modifications should address the concern that the current comparison is relatively insufficient.
>
> - ### **Limited Literature Review** (Reviewers cNCZ , efQf, and 683i)
>
> **We have updated the Related Work section** with an additional paragraph covering RL-based motion planning to ensure our contribution is clearly positioned:
>
> >> Deep reinforcement learning (DRL) has been widely explored in mobile robot navigation [2]. However, these works are not directly applicable to the parking task we study in this work. Most navigation methods [3,4,5, 6] assume differential-drive robots with highly flexible motion capabilities, whereas parking requires vehicle modeling governed by nonholonomic constraints such as the bicycle or the Ackermann-steering models. These kinematic models restrict maneuverability. A related study uses an RC-car platform and combines model-free and model-based RL for indoor navigation [7]. Despite these efforts, navigation goals are typically treated as waypoints without enforcing precise final orientation, while parking requires exact terminal conditions. To the best of our knowledge, few learning-based methods jointly consider these constraints, motivating the development of our RL-based parking planner.
>
> We now clearly highlight a key distinction emphasized in the reviews: our task requires nonholonomic, reverse-gear, centimeter-accurate terminal pose constraints. This clarification more precisely positions our contributions relative to existing RL literature.
>
> - ### **Modest Novelty and Contribution** (Reviewers efQf and 683i)
>
> Reviewers efQf and 683i questioned the novelty and contribution framing. **We have updated the writing** and strengthened the narrative by emphasizing the following unique aspects:
>
> (1) ParkBench is the first benchmark for real parking layouts, designed to evaluate constrained, multi-maneuver parking in a reproducible way.
>
> (2) Our method integrates several practical and non-trivial components, including roll-out feasibility filtering, nonholonomic curriculum design, and action-chunking wrapper for long-horizon generation. We achieve state-of-the-art performance in tight environments.
>
> (3) We added further explanation on how the proposed framework supports real-vehicle deployment due to its closed-loop nature and tracking-friendly path outputs.
>
> We believe these revisions address the concerns raised by reviewers efQf and 683i regarding the clarity of our contributions.

---

> ### Author Response · Authors · 2025-11-26
> **Rebuttal Summary (Continued)**
>
> Again, we thank the reviewers for identifying opportunities to strengthen the paper. Their feedback has significantly improved the clarity and completeness of our work. We also thank the Area Chair for carefully tracking the rebuttal process and for their efforts throughout the review cycle. We hope the revisions demonstrate that the concerns are fully addressable and that the combination of a new real-world benchmark and a practical, high-performing RL-based parking planner offers a meaningful contribution to the ICLR community. We hope the revisions demonstrate that the concerns are addressable and that the combination of a new real-world benchmark and a practical, high-performing RL-based parking planner offers a meaningful contribution to the ICLR community.
>
> ### Reference:
>
> [1] Zhang, P., Zhou, S., Hu, J. et al. Automatic parking trajectory planning in narrow spaces based on Hybrid A* and NMPC. Sci Rep 15, 1384 (2025). https://doi.org/10.1038/s41598-025-85541-x
>
> [2] Zhu, Kai, and Tao Zhang. "Deep reinforcement learning based mobile robot navigation: A review." Tsinghua Science and Technology 26.5 (2021): 674-691.
>
> [3] C. Pérez-D’Arpino, C. Liu, P. Goebel, R. Martín-Martín and S. Savarese, "Robot Navigation in Constrained Pedestrian Environments using Reinforcement Learning," 2021 IEEE International Conference on Robotics and Automation (ICRA), Xi'an, China, 2021, pp. 1140-1146
>
> [4] X. Ruan, D. Ren, X. Zhu and J. Huang, "Mobile Robot Navigation based on Deep Reinforcement Learning," 2019 Chinese Control And Decision Conference (CCDC), Nanchang, China, 2019, pp. 6174-6178
>
> [5] Xu, Zifan, et al. "Benchmarking reinforcement learning techniques for autonomous navigation." arXiv preprint arXiv:2210.04839 (2022).
>
> [6] G. Kahn, A. Villaflor, B. Ding, P. Abbeel and S. Levine, "Self-Supervised Deep Reinforcement Learning with Generalized Computation Graphs for Robot Navigation," 2018 IEEE International Conference on Robotics and Automation (ICRA), Brisbane, QLD, Australia, 2018, pp. 5129-5136
>
> [7] Akmandor, Neşet Ünver, et al. "Deep reinforcement learning based robot navigation in dynamic environments using occupancy values of motion primitives." 2022 IEEE/RSJ international conference on intelligent robots and systems (IROS). IEEE, 2022.

---

### Meta-Review · Area_Chair_zap1 · 2026-01-06

**Summary:**

Reviewer cNCZ:
- remaining concerns are shared with other reviewers (https://openreview.net/forum?id=T98uLLyWiM&noteId=f2DdKapKb1)

Reviewer efQf:
- method is straightforward and well-established by prior work
- missing comparison to prior works
- speculative claims

Reviewer 683i:
- misclaim of novelty of method
- comparison to a single classical baseline
- missing comparisons on additional tasks

Reviewer JmMd:
- all experiments in static environments

**Reviewer Concerns:**

Outstanding concerns:
- missing comparisons to additional classical baselines
- missing comparisons on additional tasks
- all experiments in static environments

Overall, there are a variety of weaknesses in the evaluation that undermine the paper's novelty. The claims are mostly accurate but together incremental.

**Reviewer Scores:**

- Reviewer cNCZ: 6->6
- Reviewer efQf: 2->4
- Reviewer 683i: 2->2
- Reviewer JmMd: 4->4

---

### Decision · Program_Chairs · 2026-01-26

Reject